# Night-Time Hot Spring Bathing Is Associated with a Lower Systolic Blood Pressure among Japanese Older Adults: A Single-Institution Retrospective Cohort Study

**DOI:** 10.3390/geriatrics9010002

**Published:** 2023-12-21

**Authors:** Satoshi Yamasaki, Tomotake Tokunou, Toyoki Maeda, Takahiko Horiuchi

**Affiliations:** 1Department of Internal Medicine, Kyushu University Beppu Hospital, Beppu 874-0838, Japan; tokunout@fdcnet.ac.jp (T.T.);; 2Department of Hematology and Clinical Research Institute, National Hospital Organization Kyushu Medical Center, Fukuoka 810-0065, Japan; 3Division of Basic Medical Science and Fundamental Nursing, Department of Nursing, Fukuoka Nursing College, Fukuoka 814-0193, Japan

**Keywords:** hypertension, hot spring bathing, older adult, night time

## Abstract

Hot spring bathing is practiced to help manage hypertension. We retrospectively investigated the effects of hot spring bathing on hypertension with the aim of identifying a novel approach to prevent and manage hypertension. The study cohort comprised 99 patients aged ≥65 years admitted to Kyushu University Beppu Hospital between 1 December 2021 and 30 November 2022 who could walk by themselves and who used hot springs for ≥3 days during their hospital stay. The changes in both systolic and diastolic blood pressure were significantly decreased in the night-time bathing group (n = 21) compared with the noontime (n = 26) and afternoon (n = 52) groups. Night-time hot spring bathing was significantly associated with reduced systolic blood pressure the next morning in older adults. Although prospective randomized controlled trials on night-time hot spring bathing as a hypertension treatment are warranted to investigate whether the practice can prevent hypertension among adults aged ≥65 years, we have initiated a single-center, phase II study on the relationship between sleep quality and quality of life in hypertensive patients after night-time hot spring bathing.

## 1. Introduction

Hypertension is the leading reason for hospital visits and long-term medication use [1,2,3]. In Japan, the prevalence of hypertension is ≥60% among men aged ≥50 years and women aged ≥60 years according to a national survey over a 55-year period [4]. The treatment of hypertension should involve lifestyle modifications, such as dietary salt restriction [5], weight loss [6], diet [7], exercise [8], and limited alcohol intake [3].

The use of hot spring bathing has increased in Japan and other Asian countries. Hot spring bathing seems to have several preventive effects on various diseases, including hypertension [9]. The high temperature of hot spring baths dilates the blood vessels, which lowers blood pressure, increases the volume of blood pumped by the heart [10], and effects cardiac output and heart rate, as these are linked to blood pressure changes [11]. Stress has two components, an acute phase and a chronic phase [12]. There is a tendency for psychosocial stresses to cluster together; this leads to a substantially increased risk of cardiac events that is equal to that of previously identified risk factors for cardiovascular disease, such as hypertension [13]. Pinheiro et al. have reviewed the main enzymatic sources of oxidants and the main antioxidant defense systems in the vascular wall. They discuss the role of oxidative stress in hypertension, which includes processes of endothelial dysfunction, vascular remodeling, and tissue damage [14]. Recently, Yamasaki et al. [15] reported an inverse relationship between habitual night-time hot spring bathing and a lower prevalence of hypertension. However, the details of the relationship between hot spring bathing and hypertension remain unknown.

To address the knowledge gap regarding the management of older patients with hypertension, we retrospectively examined the relationship between hot spring bathing and changes in blood pressure associated with hot spring bathing in hospitalized older patients. We identified a cohort of patients admitted to our institution for whom there was a complete dataset on blood pressure measurements before and after hot spring bathing and who were ambulatory without assistance. We identified dependent variables according to our prespecified hypothesis of an inverse correlation between night-time hot spring bathing and blood pressure, as reported in a previous cross-sectional study [15].

## 2. Methods

### 2.1. Study Population

The study population included 240 patients aged ≥65 years who were admitted to the Department of Internal Medicine of Kyushu University Beppu Hospital in Beppu between 1 December 2021 and 30 November 2022, with a primary or most responsible diagnosis of cardiovascular, collagen, hematological, or oncological diseases. Using the hospital medical records database, we retrospectively reviewed the data of patients who could walk by themselves (so that the timing of bathing reflected the patients’ preferences) and who used hot springs for 3 days or more (because almost all inpatients who used hot springs were in hospital for at least 3 days). The use of hot springs was limited for inpatients to between 12:00 and 20:00 once per day of hospitalization. These limitations were imposed because patients’ vital signs, such as blood temperature, heart rate, and blood pressure, were measured and recorded by the nurses in the morning and 1–2 h after breakfast and taking medication, which is the standard of care for inpatients at our institution. The data, which included hot spring bathing time, were recorded in a consistent and reliable format in the computerized medical records system organized by experts. All inpatients were prescribed a special diet organized by nutritional specialists. Some components of this diet, such as reduced salt intake, were potential confounders for blood pressure and were therefore considered as potential effect modifiers for the reduction in blood pressure. We excluded patients whose morning blood pressure before and after hot spring bathing was measured during invasive tests, procedures, or therapies, including those who changed antihypertensive medication during the hospital stay and who needed physical therapy, which was considered a potential confounder for blood pressure. All data were collected and double-checked for accuracy by two separate research assistants. The data included age, sex, and disease history. The examined variables were as follows: age; sex; hypertension with medication; and lifetime disease history, including benign cardiac arrhythmia (i.e., arrhythmias that are not associated with symptoms or hemodynamic instability and that have no prognostic significance), stroke, gout, diabetes mellitus, hyperlipidemia, renal disease, and chronic hepatitis, which are associated with a lower prevalence of hypertension [15]. Informed consent for study participation was obtained by providing participants with information on our hospital website. This study was performed in accordance with the institutional guidelines and the principles of the Declaration of Helsinki. The protocol was approved by the institutional review board of Kyushu University Hospital, Japan (No. 23020-00).

### 2.2. Statistical Methods

We analyzed the frequencies and descriptive statistics for the participant data. We excluded all patients who had missing data for at least one of the covariates. Intergroup differences in categorical variables are expressed as numbers and percentages. The chi-square test, which is appropriate for categorical data and for nonparametric distributions, was used to examine the relationships between the categorical variables. To evaluate the association between hot spring bathing time and changes in blood pressure, we compared the changes in systolic and diastolic blood pressure among the noontime (12:00 to 13:00), afternoon (13:00 to 19:00), and night-time (19:00 to 20:00) hot spring bathing groups. *p* values of <0.05 were considered statistically significant. All analyses were conducted using EZR (Saitama Medical Center, Saitama, Japan; http://www.jichi.ac.jp/saitama-sct/SaitamaHP.files/statmedEN.html, accessed on 11 April 2022) [16], which is a graphical user interface for R version 2.13.0 (www.r-project.org), and a modified version of R Commander version 1.6–3 designed to add statistical functions.

## 3. Results

Overall, 99 of 240 patients (41.2%) met all of the eligibility criteria and underwent hot spring bathing around noon (n = 26), in the afternoon (n = 52), or at night (n = 21) three times or more at our institute. The baseline characteristics of the patients who used hot springs around noon, in the afternoon, and at night are shown in Table 1. The mean (interquartile range) age of the patients who used hot springs around noon, in the afternoon, and at night, respectively, was 72 (5), 79 (16), and 78 (14) years. The patients in the noontime group were younger than the patients in the afternoon and night-time groups (***p*** = 0.0036; chi-square test). In the noontime, afternoon, and night-time groups, 23%, 56%, and 71% of patients, respectively, were female. There were more female patients in the night-time group than in the noontime and afternoon groups (***p*** = 0.0019; chi-square test). The diastolic blood pressure at baseline for all patients was <90 mmHg (the mean baseline diastolic blood pressure was 61–80 mmHg).

To evaluate the association between hot spring bathing time and changes in blood pressure, we compared the changes in systolic and diastolic blood pressure in the morning between the noontime, afternoon, and night-time groups (Figure 1a,b; chi-square test). The changes in both systolic and diastolic blood pressure were significantly decreased in the night-time bathing group compared with the noontime and afternoon groups. In addition, although the systolic blood pressure decreased, the diastolic blood pressure did not change after night-time hot spring bathing. In both the noontime and afternoon groups, the systolic and diastolic blood pressure was higher in the morning before hot spring bathing than in the morning after hot spring bathing (Figure 2a,b and Figure 3a,b; chi-square test). In the night-time group, the systolic blood pressure was lower in the morning before hot spring bathing than in the morning after hot spring bathing, but there was no significant difference in the diastolic blood pressure in the morning before versus after hot spring bathing (Figure 2c and Figure 3c; chi-square test).

## 4. Discussion

In this single-institution, retrospective cohort study, the changes in both the systolic and diastolic blood pressure were significantly decreased in the night-time bathing group compared with the noontime and afternoon groups. In addition, the systolic blood pressure decreased, but the diastolic blood pressure did not change after night-time hot spring bathing. Both the systolic and diastolic blood pressure after noontime and afternoon hot spring bathing were increased.

To date, only one study has investigated the association between habitual night-time hot spring bathing and a lower prevalence of hypertension [15]. The present study showed, for the first time, that night-time hot spring bathing decreases both the systolic and diastolic blood pressure in the morning, especially systolic blood pressure. Tai et al. [17] reported that Japanese hot water bathing, especially in the short time from the end of bathing to bedtime, was associated with lower night- and sleep-time blood pressure and greater dipping in an older adult population. Ishikawa et al. [18] found a reduction in the systolic and diastolic blood pressure after hot water bathing, but there was no difference in the night-time blood pressure between bathing days and non-bathing days. In the present study, we selected patients who had blood pressure measurements taken in the morning because blood pressure varies according to many internal and external factors. For example, behavioral factors play an important role in diurnal blood pressure variation, and blood pressure rises sharply on waking in the morning [19]. In a study by Obara et al. [20], more than half of the treated patients were classified as having uncontrolled morning hypertension, and these patients had a poorer prognosis. In an American study, the control of blood pressure to a systolic pressure of <130 mmHg and a diastolic pressure of <80 mmHg after antihypertensive drug therapy was achieved by 54%, 50%, 46%, and 33% of patients aged 20–54 years, 55–64 years, 65–74 years, and ≥75 years, respectively [2]. Additionally, 60%–80% of older patients with hypertension have isolated systolic hypertension, with a systolic blood pressure of >160 mmHg and a diastolic blood pressure of <90 mmHg [21]. According to one previous study, systolic blood pressure is a major predictor of coronary artery disease in older patients [22]. Elevated systolic blood pressure is mainly caused by reduced arterial compliance, and isolated systolic hypertension may result from an increase in cardiac output owing to anemia, aortic insufficiency, or arteriovenous fistula [23]. Therefore, night-time hot spring bathing to reduce systolic blood pressure may be more suitable for older adults than for younger adults.

Several studies have suggested that lower diastolic pressure is associated with an increased risk of coronary heart disease [21,24,25,26]. Our observation of an increase in diastolic blood pressure after noontime and afternoon hot spring bathing may indicate that such activity has a positive effect for older patients in terms of reducing the risk of coronary heart disease. Understanding the effect of hot spring bathing on reducing the risk of coronary heart disease is important and requires further research. However, as diastolic blood pressure at baseline in this study was <90 mmHg, we cannot confirm that the antihypertensive effect was associated with the use of night-time hot springs.

Previous studies have indicated the utility of traditional thermal therapy, including hot spring bathing, for hypertension [27,28,29]. However, such studies have used small sample sizes. Of course, it is possible that hypertension affects skin blood flow and sweating [30], which prevent persistent elevation in the core temperature by facilitating whole-body heat loss during hot spring bathing. In the present study, we found that night-time hot spring bathing, which can improve sleep disorders, may be inversely associated with changes in systolic and diastolic blood pressure in older patients. In a large-scale study of an older population, night-time hot spring bathing was associated with shorter sleep onset latency when bathing was scheduled 1–3 h before bedtime, and with a higher distal–proximal skin temperature gradient when bathing took place 30 min before bedtime [31]. At our institute, the bedtime was 21:00; therefore, patients in the night-time group used hot spring bathing 1–2 h before bedtime. Sawatari et al. [32] suggested that leg thermal therapy could improve subjective and objective sleep quality in patients with chronic heart failure. It is therefore possible that night-time hot spring bathing improves sleep, which may improve hypertension control [33].

Many risk factors, including age, obesity, family history, high-sodium diet, and physical inactivity, are associated with hypertension development [7,8]. Spending on adult hypertension in the United States from 1996 to 2016 increased by USD 70 billion [34]. In the current context of population aging and population growth, there is substantial interest among clinicians and researchers in developing proactive and preventive interventions as alternatives to reactive approaches to hypertension. In this study, we demonstrated that an alternative option for potentially improving hypertension control in older adults is habitual night-time hot spring bathing. A previous study indicated that repeated Finnish sauna use substantially reduces cortisol concentrations [35], which may cause low blood pressure. Different results regarding hormone responses are likely owing to differences in study methods and consideration of factors such as the duration, time, and frequency of bathing. Immersion while bathing can cause the release of atrial natriuretic peptide, which is a cardiac hormone that regulates blood pressure and volume [36]. Such factors should be considered in long-term observational studies. Understanding the cardiovascular response would provide a more comprehensive account of the physiological response to hot spring bathing. Because there is a lack of physical activity and healthy nutrition among older patients, practical interventions to prevent or improve hypertension, such as habitual night-time hot spring bathing, warrant additional attention.

The present study has some limitations that should be acknowledged. As this was a retrospective cohort study performed at a single institution, there was a possibility of selection bias, poor control over external variables and covariates, and potential confounders. Statistical analysis is only appropriate for quantitative results. One limitation of the chi-square test, which we used, is that it cannot establish the potential causality of any of the associations. However, the chi-square test is considered a robust test. In this study, bias was present owing to differences in data selection and other factors, including lack of data regarding patients who engaged in hot spring bathing for the treatment of various diseases; patients’ income, which might correlate with some vascular diseases or the frequency of hot spring bathing; patients’ lifestyle and diet, including food consumption and obesity, sodium intake, drinking and smoking habits, consumption of coffee and tea, physical activity, and sleep; and the likelihood of patients engaging in hot spring bathing at a particular time. To minimize bias, we limited the inclusion criteria to patients with data on age, sex, disease history, and use of hot spring bathing at our institute during administration. Second, no long-term data on blood pressure were available; thus, additional studies are needed to perform more detailed analyses of the long-term outcomes of patients who undergo hot spring bathing. Third, patients with hypertension may have been overlooked or the changes in blood pressure underestimated because of antihypertensive drug therapy. Moreover, there were no data specific to hot spring bathing, such as the duration of immersion, frequency, temperature, time, and years of habitual hot spring bathing. In addition, there was no real control group (i.e., patients who did not engage in bathing). Furthermore, the obtained sample was small because we obtained data from the medical records of a single institution. Finally, the purpose of this study was to help clarify the relationship between hypertension prevention and habitual hot spring bathing. However, it was difficult to interpret the obtained evidence because of the lack of data on the bathing-specific aspects mentioned above (e.g., temperature, immersion duration) and because we could not retrospectively evaluate the quality of the medical data.

## 5. Conclusions

In this study, night-time hot spring bathing in older adults was significantly associated with reduced systolic blood pressure compared with noontime and afternoon bathing. It is important to prioritize clinical trials on the prevalence and treatment of hypertension during hot spring bathing, such as how the timing and frequency of bathing differentially affects blood pressure, the effects of bathing-related factors including temperature and immersion duration, and how applicable the use of habitual bathing for patients in cultures that do not have such a tradition is. Although our findings suggest the benefits of hot spring bathing for hypertension, more work is needed to clarify the specific mechanisms underlying this effect and to determine which mechanisms are more important. At the beginning, we need to examine if hot spring bathing is to be recommended for hypertension prevention as well as hypertension control, how feasible it is in terms of access, cost, and time for people with hypertension to engage in regular hot spring bathing, and if there are any standard guidelines available for the use of this method to prevent or control hypertension. Although randomized controlled trials on habitual night-time hot spring bathing as a treatment for hypertension are warranted, we have initiated a single-center, phase II study on the relationship between sleep quality and quality of life in hypertensive patients after night-time hot spring bathing. This protocol was approved by the Institutional Review Board of Kyushu University Hospital, Japan (No. 20232002).

## Figures and Tables

**Figure 1 geriatrics-09-00002-f001:**
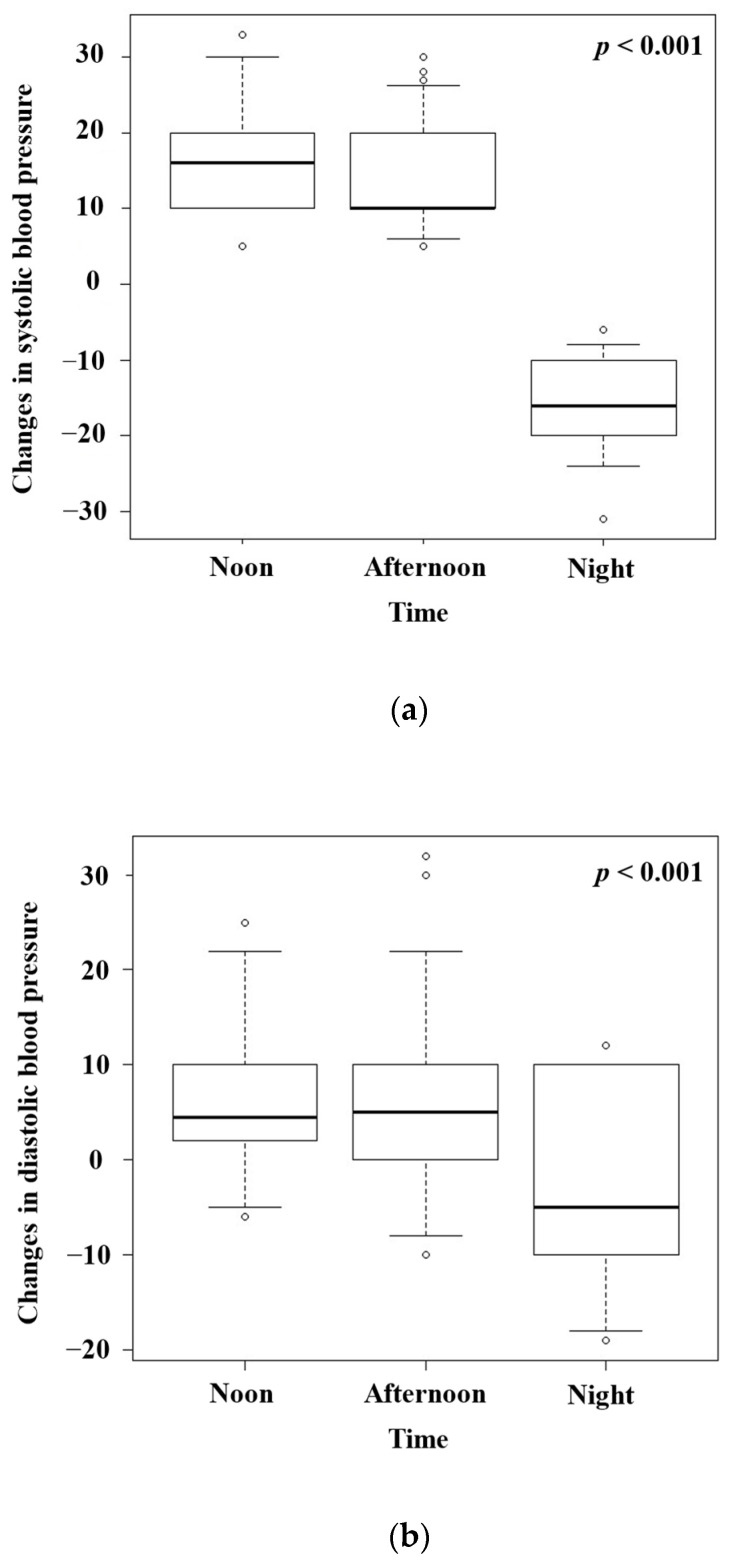
(**a**) Changes in systolic blood pressure in the noontime, afternoon, and night-time hot spring bathing groups. (**b**) Changes in diastolic blood pressure in the noontime, afternoon, and night-time hot spring bathing groups.

**Figure 2 geriatrics-09-00002-f002:**
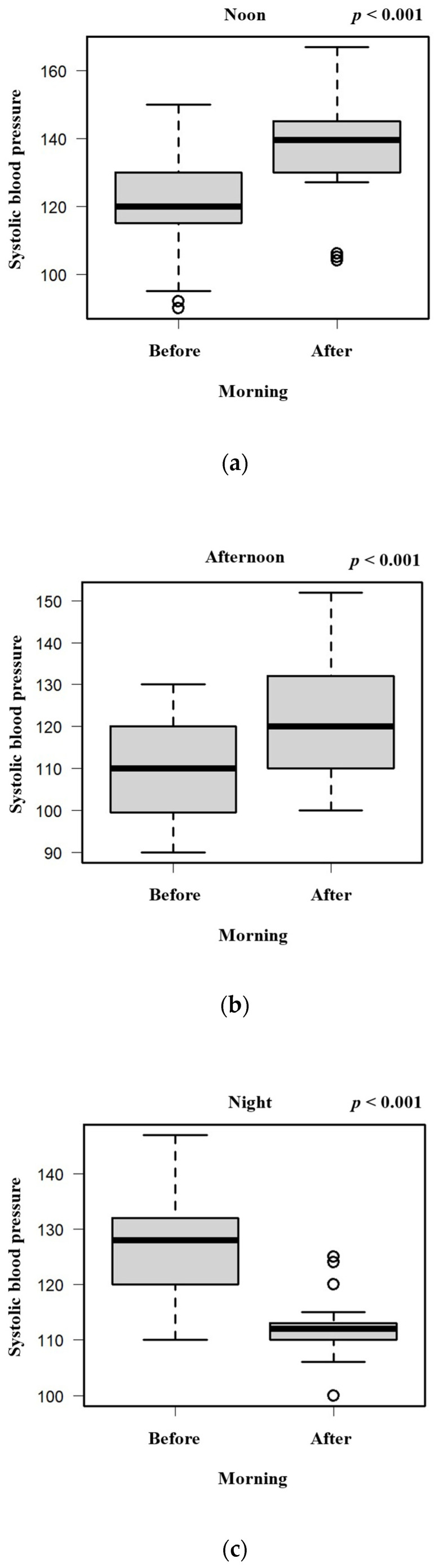
Systolic blood pressure in the morning before and after noontime (**a**), afternoon (**b**), or night-time (**c**) hot spring bathing.

**Figure 3 geriatrics-09-00002-f003:**
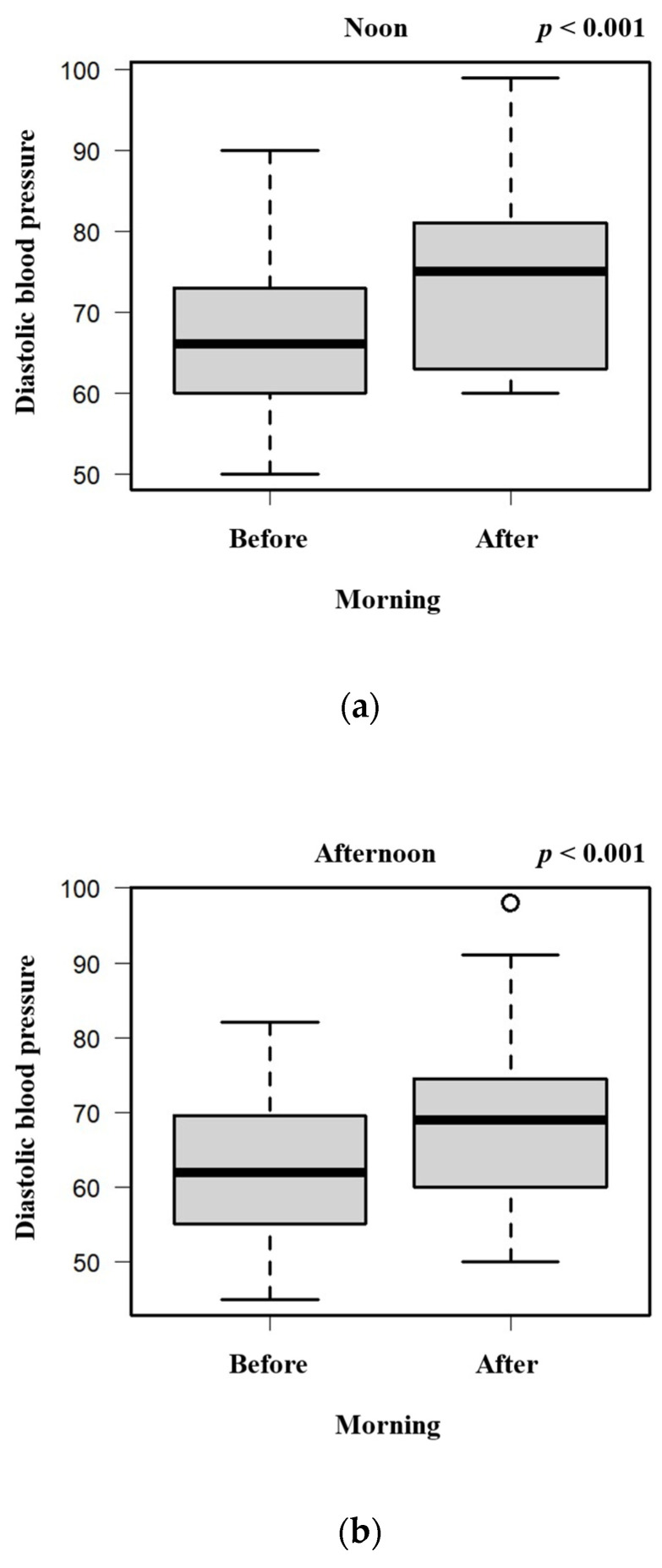
Diastolic blood pressure in the morning before and after noontime (**a**), afternoon (**b**), or night-time (**c**) hot spring bathing.

**Table 1 geriatrics-09-00002-t001:** Patients’ characteristics.

	Time			Baseline BP	
Characteristics	Noon	Afternoon	Night	SBP	DBP
	n = 26	n = 52	n = 21	Mean (IQR), mmHg	
**Age, mean (IQR), year**	72 (5)	79 (16)	78 (14)		
**Female (%)**	6 (23)	29 (56)	15 (71)		
**Comorbidity (%)**					
**Hypertension**	9 (35)	11 (21)	9 (43)	114 (15)	67 (12)
**Benign cardiac arrhythmia**	0	3 (6)	3 (14)	112 (2)	61 (11)
**Stroke**	3 (12)	0	0	132 (10)	80 (9)
**Gout**	0	3 (6)	0	122 (2)	66 (5)
**Diabetes mellitus**	9 (35)	23 (44)	9 (43)	116 (13)	64 (10)
**Hyperlipidemia**	6 (23)	11 (21)	3 (14)	120 (14)	70 (9)
**Renal disease**	0	7 (14)	2 (10)	117 (8)	66 (9)
**Chronic hepatitis**	3 (12)	0	0	92 (2)	61 (1)

IQR, interquartile range; BP, blood pressure; SBP, systolic BP; DBP, diastolic BP.

## Data Availability

We used data obtained from patients’ medical records at Kyushu University Beppu Hospital, Beppu, Japan. The datasets generated and/or analyzed during the current study are not publicly available owing to privacy and confidentiality restrictions pertaining to personal health information. However, the dataset creation plan is available from the corresponding author on reasonable request.

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
