# Peer review of "Night-Time Hot Spring Bathing Is Associated with a Lower Systolic Blood Pressure among Japanese Older Adults: A Single-Institution Retrospective Cohort Study"

_geriatrics, 2023, doi:10.3390/geriatrics9010002_

Round 1
Reviewer 1 Report
Comments and Suggestions for Authors
This study shows that nighttime hot spring bathing is associated with lower systolic blood pressure in Japanese older adults in a single center. It is interesting, but as the author describes in the limitation, it is difficult to understand the bias or interpretation of the effect of night-time hot spring bathing from the study.
1.In the abstract, "Female sex and arrhythmia were independently and significantly associated with nighttime hot spring bathing". The reviewer could not understand why the author described in the abstract. Arrhythmia was a broad word. Moreover, does it mean that the arrhythmia was probably caused by night-time hot spring bathing?
2. There is only one table and one figure. Reviewers think that some or all of the supplementary figures and tables must be better or more understandable to Table2,3 or Figure 2,3 etc. In addition, for example, Figure 1 a-c are better on the same slide rather than on separate slides. If the author changes like this, reader can compare blood pressure 3 pattern hot spring bath timing within one slide.
3. In Table 1, underlying disease, baseline systolic or diastolic blood pressure are required. P value are required. On the other hand, arrythmia doesn't need in the list because of wide (from observation to lethal arrythmia).
4. How do patients separate noon, afternoon and night? Female was significantly independent factor of night bathing. This is retrospective data, so the reviewer judge that the patient selected noon, afternoon, night. If so, there was no relationship of sex for the effect of blood pressure for night bathing.
5. The reviewer cannot see the supplementary tables. However, in our opinion, there is no need for supplementary Fg or tables. If possible, I recommend that all tables or figures should be listed in the text without supplement.
6. If the main data is OR for night-time hot spring bathing, the author should list the univariate and multivariate analysis as a table.
7. P6, "After the end of sauna bathing, blood pressure decreased steadily compared with before sauna bathing" There is no relation of sauna bathing in this study, so the reviewer feels that the related sentences could be deleted. The reader may be confused and misunderstood.
Author Response
This study shows that nighttime hot spring bathing is associated with lower systolic blood pressure in Japanese older adults in a single center. It is interesting, but as the author describes in the limitation, it is difficult to understand the bias or interpretation of the effect of night-time hot spring bathing from the study.
1.In the abstract, "Female sex and arrhythmia were independently and significantly associated with nighttime hot spring bathing". The reviewer could not understand why the author described in the abstract. Arrhythmia was a broad word. Moreover, does it mean that the arrhythmia was probably caused by night-time hot spring bathing?
RESPONSE: We apologize for the unclear text in the Abstract. Our meaning was that female patients and patients with benign cardiac arrhythmia showed a preference for night-time hot spring bathing. We have revised the Abstract to clarify this (page 1, lines 23–25).
- There is only one table and one figure. Reviewers think that some or all of the supplementary figures and tables must be better or more understandable to Table2,3 or Figure 2,3 etc. In addition, for example, Figure 1 a-c are better on the same slide rather than on separate slides. If the author changes like this, reader can compare blood pressure 3 pattern hot spring bath timing within one slide.
RESPONSE: In accordance with your suggestion, we have moved the data presented in the Supplementary Table and Figures to the main tables and figures. Supplementary Table 1 is now Table 2. Supplementary Figure 1 is now Figure 2. Supplementary Figure 2 is now Figure 3 (page 3, lines 126 and 129; page 7, line 160).
- In Table 1, underlying disease, baseline systolic or diastolic blood pressure are required. P value are required. On the other hand, arrythmia doesn't need in the list because of wide (from observation to lethal arrythmia).
RESPONSE: As requested, we have added data for baseline systolic and diastolic blood pressure to Table 1. We were unable to perform statistical analysis on these data because some patients had multiple diseases. All patients with arrythmia in this study had benign cardiac arrythmia, as described in the Methods (page 2, lines 77–79). We have revised “arrythmia” to “benign cardiac arrythmia” (page 1, line 23; page 2, line 77; page 7, lines 151, 153–154, 158, 175–176, 217, 220, and 284; Table 1; Table 2).
- How do patients separate noon, afternoon and night? Female was significantly independent factor of night bathing. This is retrospective data, so the reviewer judge that the patient selected noon, afternoon, night. If so, there was no relationship of sex for the effect of blood pressure for night bathing.
RESPONSE: The timing of bathing was selected according to patients’ preferences and recorded in a consistent and reliable format in the computerized medical records system, as described in the Methods (page 2, lines 77–79). We apologize for the lack of clarity in the text. Female patients preferred night-time hot spring bathing. We have revised the relevant text to explain that female sex was independently and significantly associated with a preference for night-time hot spring bathing (page 1, line 24; page 7, lines 155, 159, 162, 166, and 177; page 8, lines 212 and 218).
- The reviewer cannot see the supplementary tables. However, in our opinion, there is no need for supplementary Fg or tables. If possible, I recommend that all tables or figures should be listed in the text without supplement.
RESPONSE: In accordance with your suggestion, we have moved the data presented in the Supplementary Table and Figures to the main tables and figures. Supplementary Table 1 is now Table 2. Supplementary Figure 1 is now Figure 2. Supplementary Figure 2 is now Figure 3 (page 3, lines 126 and 129; page 7, line 160).
- If the main data is OR for night-time hot spring bathing, the author should list the univariate and multivariate analysis as a table.
RESPONSE: As described in our response to Comment 5, we have revised the Supplementary Table and Figures (page 3, lines 126 and 129; page 7, line 160). Table 2 now shows both the univariate and multivariate results.
- P6, "After the end of sauna bathing, blood pressure decreased steadily compared with before sauna bathing" There is no relation of sauna bathing in this study, so the reviewer feels that the related sentences could be deleted. The reader may be confused and misunderstood.
RESPONSE: We apologize for the unclear expression. We acknowledge your point and have removed the unclear text on sauna bathing in the Discussion.
Reviewer 2 Report
Comments and Suggestions for Authors
Abstract: "who used hot springs for 3 days or more" during their stay in the hospital.
Abstract: "Night-time hot spring bathing was significantly associated with a reduced systolic blood pressure" the next morning "in older adults."
Table 1:
the percentage for hypertension is wrong 9/35=35% and not 65%. Some other percentages in the table are also not correct.
I think even for the primary criteria of your trial, systolic and diastolic blood pressure measured the mornign after a hot bath, the sample size for a retrospective trial is quite small. (this is an important limitation).
You should better discuss, why only the systolic BP was lowered
Main comment:
I think, for any subgroup analysis the sample size is far too small and the "significant" OR you found for females an arrhythmia are probably not really singificant. You should leave this whole part away and stay with your main finding, which is quite important.
Comments on the Quality of English Language
It is clear, that the text is not from a native speaker (I myself am not a native English speaker), but it is well understandable.
Author Response
Abstract: "who used hot springs for 3 days or more" during their stay in the hospital.
RESPONSE: Thank you for your helpful suggestion. We have added this text to the Abstract (page 1, line 19).
Abstract: "Night-time hot spring bathing was significantly associated with a reduced systolic blood pressure" the next morning "in older adults."
RESPONSE: Thank you for your helpful suggestion. We have added this text to the Abstract (page 1, line 26).
Table 1:
the percentage for hypertension is wrong 9/35=35% and not 65%. Some other percentages in the table are also not correct.
RESPONSE: We apologize for the error in Table 1. We have corrected the data in the revised table.
I think even for the primary criteria of your trial, systolic and diastolic blood pressure measured the mornign after a hot bath, the sample size for a retrospective trial is quite small. (this is an important limitation).
RESPONSE: Thank you for your helpful suggestion. We agree that this study had several limitations, including the small sample size. We have added a discussion of this as a study limitation (page 9, lines 273–275).
You should better discuss, why only the systolic BP was lowered
RESPONSE: Thank you for raising this issue. We agree with your comment and have added a mention of this point to the Discussion (page 8, lines 208–210).
Main comment:
I think, for any subgroup analysis the sample size is far too small and the "significant" OR you found for females an arrhythmia are probably not really singificant. You should leave this whole part away and stay with your main finding, which is quite important.
RESPONSE: We apologize for the unclear text. Female patients and patients with benign cardiac arrhythmia showed a preference for night-time hot spring bathing. We have revised the text to clarify this (page 1, line 24; page 7, lines 155, 159, 162, 166, and 177; page 8, lines 212 and 218).
Round 2
Reviewer 1 Report
Comments and Suggestions for Authors
There are no further comments.
Author Response
Thank you for your thoughtful and constructive comments on our manuscript; we appreciate your help.
Reviewer 2 Report
Comments and Suggestions for Authors
Significance is only valid for one primary criterium (morming blood pressure), or you have to use methods for multiple testing. I still believe strongly, that your OR in the multivariate analysis are still pure chance.
So just don`t use this word (not in line 24, 95, 98, 155, 159, ...) if it does not correspond to the blood pressure.
the part in the discussion lone 210 -221 I would just leave away.
In the lmitations you have to add , that there is no real control group (not bathing)
I could not fion this study in your introduction or discussion
Ishikawa, J., Yoshino, Y., Watanabe, S., & Harada, K. (2016). Reduction in central blood pressure after bathing in hot water. Blood Pressure Monitoring, 21(2), 80-86.
In the discusson it would be nice if you mention the different possible mechnisms as temperature of the water and the pressure effects of immersin in water (resulting in production of ANP and the diuresis)
For example in line 95: Covaraites that had a p<0.05 in the univariate ...
Itis good that the figures are now in the article
Comments on the Quality of English Language
line 37 instead of "uptake" better "use"
Author Response
Significance is only valid for one primary criterium (morning blood pressure), or you have to use methods for multiple testing. I still believe strongly that your OR in the multivariate analysis is still pure chance.
So just don`t use this word (not in line 24, 95, 98, 155, 159, ...) if it does not correspond to the blood pressure.
the part in the discussion lone 210 -221 I would just leave away.
RESPONSE: In accordance with your suggestion, we have removed the text on the multiple regression analysis (including the odds ratio and confidence interval values) and have removed Table 2. Therefore, in the revised manuscript, significance is mentioned only in relation to blood pressure. We have also removed the text you refer to in lines 210–221 of the Discussion.
In the limitations you have to add, that there is no real control group (not bathing)
RESPONSE: As requested, we have added this factor as a limitation in the Discussion (page 8, lines 235–236).
I could not find this study in your introduction or discussion.
Ishikawa, J., Yoshino, Y., Watanabe, S., & Harada, K. (2016). Reduction in central blood pressure after bathing in hot water. Blood Pressure Monitoring, 21(2), 80-86.
RESPONSE: In accordance with your suggestion, we have added a citation to this study to the Discussion (page 7, lines 151–153).
In the discussion it would be nice if you mention the different possible mechanisms as temperature of the water and the pressure effects of immersion in water (resulting in production of ANP and the diuresis)
RESPONSE: Thank you for raising this issue. We agree with your comment and have added a mention of these factors to the Discussion (page 7, lines 181–183 and page 8, lines 211–213).
For example, in line 95: Covariates that had a p<0.05 in the univariate ...
RESPONSE: As the multiple regression analysis results have been omitted, this text has been removed.
It is good that the figures are now in the article
RESPONSE: Thank you for your constructive comments on our manuscript.
Comments on the Quality of English Language
line 37 instead of "uptake” better “use"
RESPONSE: We apologize for this error in the text. We have changed this word in the revised manuscript (page 1, line 34).